# The Effectiveness of Cancer Immune Checkpoint Inhibitor Retreatment and Rechallenge—A Systematic Review

**DOI:** 10.3390/cancers15133490

**Published:** 2023-07-04

**Authors:** Adrian Perdyan, Bartosz Kamil Sobocki, Amar Balihodzic, Anna Dąbrowska, Justyna Kacperczyk, Jacek Rutkowski

**Affiliations:** 13P-Medicine Laboratory, Medical University of Gdansk, 80-210 Gdansk, Poland; 2Department of Biology, Stanford University, Stanford, CA 94305, USA; 3Student Scientific Circle of Oncology and Radiotherapy, Medical University of Gdansk, 80-210 Gdansk, Poland; 4Division of Oncology, Department of Internal Medicine, Comprehensive Cancer Center Graz, Medical University of Graz, 8036 Graz, Austria; 5BioTechMed-Graz, 8010 Graz, Austria; 6The University Clinical Centre in Gdansk, Medical University of Gdansk, 80-210 Gdansk, Poland; 7Department of Oncology and Radiotherapy, Medical University of Gdansk, 80-210 Gdansk, Poland

**Keywords:** immune checkpoint inhibitors, retreatment, rechallenge, oncology, melanoma

## Abstract

**Simple Summary:**

This systematic review gathered recent findings on immune checkpoint inhibitor retreatment or rechallenge in order to overcome primary resistance. The systematic review was performed according to PRISMA and PICO guidelines. In total, 31 articles were included with a total of 812 cancer patients. There were 16 retreatment and 13 rechallenge studies. Fifteen studies reported improvement or maintenance of overall response or disease control rate at the secondary treatment. Interval treatment, primary response to immune checkpoint inhibitors and the cause of cessation from the primary immune checkpoint inhibitor therapy seemed to be promising predictors of secondary response to immune checkpoint inhibitors.

**Abstract:**

Despite a great success of immunotherapy in cancer treatment, a great number of patients will become resistant. This review summarizes recent reports on immune checkpoint inhibitor retreatment or rechallenge in order to overcome primary resistance. The systematic review was performed according to PRISMA (Preferred Reporting Items for Systematic Reviews and Meta-Analyses) guidelines. The search was performed using PubMed, Web of Science and Scopus. In total, 31 articles were included with a total of 812 patients. There were 16 retreatment studies and 13 rechallenge studies. We identified 15 studies in which at least one parameter (overall response rate or disease control rate) improved or was stable at secondary treatment. Interval treatment, primary response to and the cause of cessation for the first immune checkpoint inhibitors seem to be promising predictors of secondary response. However, high heterogeneity of investigated cohorts and lack of reporting guidelines are limiting factors for current in-depth analysis.

## 1. Introduction

The wide introduction of immunotherapy in oncology has significantly changed the prognosis and quality of life of selected cancer patients [1]. The three most commonly registered and used drugs are immune checkpoint inhibitors (ICIs) targeting programmed cell death 1 protein (PD-1), programmed death 1 ligand (PD-L1) and cytotoxic T-lymphocyte-associated protein 4 (CTLA-4) [2]. PD-1 and CTLA-4 are receptors located on T cells which may be bound by their upregulated cognate ligands expressed on tumour cell surfaces, thus suppressing the anti-tumoral immune response. ICIs work by the blockage of their interaction that leads to restoration of cytotoxic T lymphocyte function both in the tumour microenvironment (TME) and in peripheral tissue, which contributes to better therapeutic outcomes [3,4,5]. Their efficacy was initially observed in numerous clinical trials, outperforming or augmenting previously introduced oncological regimens [6,7,8,9]. Despite promising results, it turned out that ICIs are effective only in up to 20% of cancer patients [10,11]. Also, the effect is highly associated with cancer types with the best outcomes observed among immunogenic malignancies, such as melanoma and lung cancer [12]. Moreover, multiple resistance mechanisms which limit the potential of ICIs have been described [13,14]. Therefore, there is the question of if it is advisable to readminister an immunotherapy in the case of recurrence or treatment failure. Retreatment and rechallenge strategies are, lately, drawing more attention in the context of ICI studies [15,16,17].

Retreatment is defined as a repeated treatment with the same therapeutic class following relapse after adjuvant treatment has ended [18]. It concerns metastatic or unresectable disease patients who have completed prior adjuvant therapy or discontinued adjuvant therapy due to toxicity, and patients with locoregional recurrence after adjuvant therapy who subsequently underwent reception. Rechallenge follows disease progression in patients who had clinical benefit with prior treatment for unresectable or metastatic disease. It is used in patients who have disease progression after an initial response and received an alternative intervening treatment, and patients with unresectable or metastatic melanoma who had a treatment break after responding to BRAF and MEK inhibitor therapy. This concept was first established for melanoma treatment; however, recently it is becoming more commonly used in the treatment of other malignancies as well [19,20]. Hence, in this article, we summarized the effectiveness of ICI retreatment and rechallenge in different malignancies, in the light of our own experience of reusing an anti-PD-1 antibody, pembrolizumab in central nervous system (CNV) metastatic melanoma achieving long-term survival benefit (Figure 1). This article is important due to the growing number of oncological patients treated with ICI who eventually will become resistant and require other regimens. Assessment of efficacy and safety of possible ICI retreatment or rechallenge, as well as designing effective therapeutic strategies for these patients, should be of a high priority.

## 2. Materials and Methods

### 2.1. Search Strategy

The systematic review was performed according to the PRISMA (Preferred Reporting Items for Systematic Reviews and Meta-Analyses) protocol (Figure 2) [21] and facilitated a PICO-styled (Patients, Interventions, Comparisons, Outcomes) research question (Table 1) [22]. 

### 2.2. Evidence Acquisition

On the 3rd of February 2023, we performed a search using PubMed, Web of Science and Scopus. Additionally, Cochrane reviews, Google Scholar and references of included articles were checked for adequate studies. We used the following search query: (immune checkpoint inhibitors OR immunotherapy) AND (rechallenge OR retreatment) AND cancer. Initial search returned 1256 results. All articles were independently reviewed by two researchers (AP, BS) and assessed for inclusion and exclusion criteria. After initial screening, 66 articles were chosen for full-text analysis. In total, 31 met inclusion criteria and were further analyzed. 

### 2.3. Inclusion and Exclusion Criteria

We included articles reporting the effects of cancer patient rechallenge or retreatment with ICIs, including case reports and case series. Articles were excluded if authors did not report the oncological effect of ICI administration according to RECIST [23], both at the primary administration and when rechallenged or retreated. Moreover, we excluded articles in which the exact number of rechallenged or retreated patients was not stated. Studies published as abstracts, posters or reports from conferences were excluded from the analysis. Studies in languages other than English were excluded. 

### 2.4. PICO

We included studies which met predefined PICO eligibility criteria (Table 1). 

### 2.5. Evidence Synthesis

The following information was extracted from original publications and included in Table 2: name of the first author, year of the study, number of patients, stage of disease, CNV metastasis, regimens of primary treatment, treatment between primary and secondary ICI administration, time between ICI rechallenge or retreatment, regimens of ICI rechallenge or retreatment, major outcome of the study and type of cancer treated [23]. Based on reported responses, disease control rate (DCR) and objective response rate (ORR) were calculated and summarized in Table 3, with the exception of case report studies. DCR and ORR were defined as the sum of complete response (CR), partial response (PR) and stable disease (SD), and CR and PR, respectively.

## 3. Results

In total, we included and analyzed 31 articles, of which 16 were retreatment studies, 13 were rechallenge studies, and 2 were not-specified studies. There were 26 original studies on patients’ subgroups from primary cohorts, 2 case series (≤4) and 3 case reports. There were 812 patients diagnosed with several cancers including melanoma (13 studies) [24,25,26,27,28,29,30,31,32,33,34,35,36], non-small cell lung cancer (NSCLC; 9 studies) [17,37,38,39,40,41,42,43,44], breast cancer (1 study) [45], renal cell carcinoma (1 study) [46], mesothelioma (1 study) [16], classic Hodgkin lymphoma (cHL; 1 study) [47], head and neck cancers (1 study) [48], urothelial carcinoma [49] and mixed cancers (3 studies) [15,50,51]. Additionally, we included 1 retreated melanoma patient from the Department of Oncology and Radiotherapy at the University Clinical Centre of Medical University of Gdansk (Figure 1). All studies expect one [17] included patients with III or IV stage of disease.

### 3.1. Treatment Regimens

Across included studies, patients were treated with various ICI regimens. Specifically, at the primary treatment nivolumab (17 studies) or pembrolizumab (15 studies) monotherapy was the most frequent. Collectively, most patients were retreated and rechallenged with pembrolizumab (14 studies) or nivolumab (10 studies) monotherapy. A detailed description of regimens or groups of regimens used across all studies is presented in Table 2.

### 3.2. Disease Control Rate and Overall Response Rate

Based on information provided in the included studies, we calculated DCR and ORR for primary and secondary treatment for each cohort of patients (Table 3). In the vast majority, DCRs and ORRs at retreatment and rechallenge were lower than at the initial treatment. However, there were certain studies where benefit or minor decreases in tumor size were observed at secondary treatment. Specifically, these were 7 melanoma studies [24,25,26,29,32,34,35], 3 NSCLC studies [39,42,44], 1 renal cell carcinoma [46], 1 cHL study [47], 1 head and neck cancer study [48] and 2 mixed cancer studies [15,51]. Additionally, looking at the cause of cessation of the first ICI among these studies, the most frequent ones were: completion of treatment [15,24,25,26,32,34,35], disease progression [29,44,47,48], toxicity [42,51] or mixed causes completion of treatment or disease progression [39], disease progression, toxicity or other [46].

### 3.3. Progression-free Survival and Overall Survival

In Table 4, we collected PFS and OS times for retreated and rechallenged cohorts. In all, except one melanoma [29], one NSCLC [44] and one mixed cancers study [51], retreatment or rechallenge times were inferior to the initial treatment. The median PFS and OS rates at initial treatment were 6.95 months (range: 3.7–24.4) and 21.4 months (range: 15.9–39.6), respectively. The same rates at retreatment or rechallenge were 3.14 months (range: 1.6–23.6) and 15.3 months (range: 6.5–30), respectively. Additionally, we tried to establish median rates for subgroups divided by the cause of cessation of the first ICI; however, due to the high heterogeneity between analyzed studies and missing data, we resigned from applying further statistics. 

### 3.4. Treatment Toxicity

In Table 5, we summarized the numbers of ICI adverse events (AEs) observed among retreated and rechallenged patients at both treatments. To compare the rate of AEs, we used a ratio of occurred AEs to number of patients. Hence, median AE rates were 0.69 (range: 0.17–1.63) and 0.58 (range: 0.14–1.5) for initial and retreatment or rechallenge, respectively. On the other side, median severe AE (Grade ≥ 3) rates were 0.21 (range: 0–0.67) and 0.16 (range: 0–0.83), respectively. As stated previously, we resigned from applying further statistics due to high heterogeneity and missing data. 

## 4. Discussion

Current knowledge regarding ICI retreatment and rechallenge in oncology is rather limited and it is based on the low number of conducted studies and exploratory data analysis. Hence, the optimal duration of ICI therapy, predictors of response, treatment effectiveness and its safety remain unknown [52]. To summarize the current state of knowledge in this field, we present the first systematic review on ICI retreatment and rechallenge that includes studies of various cancer types and an impressive cohort of 812 patients gathered.

Across analysed studies, the most frequently used ICIs were PD-1 inhibitors pembrolizumab and nivolumab, both at the primary and secondary treatment. Moreover, they share similar three-dimensional structures and effector mechanisms; however, pembrolizumab has higher affinity for recombinant human PD-1 than nivolumab [53,54]. The next widely used drug at the secondary treatment following previous PD-1 or PD-L1 therapy, especially in melanoma patients, was a CTLA-4 inhibitor ipilimumab. However, across five melanoma studies, such a therapeutic strategy showed questionable efficacy [28,30]. Additionally, the usage of CTLA-4 inhibitors in the retreatment setting is contrary to the previously established definition consensus [18]. In summary, ICI retreatment and rechallenge guidelines should be updated for melanoma, but also for the other cancer types in which such a therapeutic strategy is getting more prevalent. It is crucial, especially in the context of the introduction of novel immunotherapies (i.e., anti-II ligand lymphocyte activation gene-3 [LAG-3] [55,56], and tebentafusp [57,58]) which are being extensively investigated in clinical trials and showing promising results in melanoma patients.

Furthermore, the general effectiveness of ICI retreatment and rechallenge is still to be confirmed. In this article, we summarized the ORR and DCR of both primary and secondary ICI treatments for each of individual studies and reported them in Table 2. Additionally, we looked at the cause of cessation for the first ICI. However, due to the high heterogeneity and the lack of statistical values in original publications, the potential pan-cancer statistical analysis would be highly biased. As expected, the efficacy measured by ORR and DCR was lower at the secondary treatment in the majority of studies. However, we identified 15 studies in which at least one parameter has improved [26,34,44,47,51], was stable, or dropped but stayed at a relatively high level (ORR or DCR ≥ 50%) [15,24,25,29,32,35,39,42,46,48]. The first ever study which showed improved ORR or DCR was conducted among eight melanoma patients treated with pembrolizumab or nivolumab and followed by ipilimumab [26]. There was no clear information about the interval treatment in this cohort. On the other side, Kan et al. showed improved responses in four melanoma patients treated with nivolumab and followed with pembrolizumab [34]. All four patients received an interval treatment. On the other side, in our case, while recurred with metastatic lesions in CNS, we followed up the patient with the same ICI agent—pembrolizumab. We believed that long-lasting complete response to primary treatment could be a predictor of response to the secondary treatment, which was in line with previous reports [24,25,32]. Further, Kambhampati et al. showed that in relapsed or refractory cHL patients, avelumab resistance can be overcome with nivolumab or pembrolizumab administration [47]. Finally, in the cohort of various cancer types (predominantly melanoma, lung cancer and lymphoma), Simonaggio et al. showed a slightly improved ORR when using anti-PD-1 or anti-PD-L1 inhibitors. Interestingly, in eight out of ten studies in which ORR or DCR was stable or dropped (but ≥50% rate), cancer patients were retreated or rechallenged with the same ICI [15,24,25,32,35,42,46,48]. This observation, which is in accordance with our own experience with pembrolizumab, may support the administration of the same drug or the same group of drugs twice, in the case of successful and well-tolerated primary treatment. However, it is worth to point out that there could be potential biological differences between patients who never achieved a complete remission and have a relatively brief progression-free interval prior to pretreatment, versus patients with lengthy initial complete remission with late relapses.

Nevertheless, it is necessary to characterize crucial factors that may be predictive for ICI retreatment or rechallenge responses. Recently, a prior or concomitant radiotherapy [59,60], chemotherapy [61,62] or targeted therapy [63] were proposed as promising strategies to improve the efficacy of primary and secondary immunotherapy. In NSCLC patients, Niki et al., Watanabe et al. and Xu et al., showed that responders to second ICI had undergone chemoradiation, chemotherapy or targeted therapy as an interval treatment [37,41,44]. The primary response to ICI seems to be a valuable factor for predicting the retreatment or rechallenge efficacy. Despite one study reporting lack of impact of primary response [30], there were several studies showing that patients who achieved SD or performed better at primary treatment were also better responding to the secondary ICIs [28,32,33,35,36,44,46]. In contrast, Fujita et al. stated that the failure of atezolizumab retreatment might be due to receiving it as a third- or later-line regimen [39], and the effectiveness is more related to PD-L1 expression rather than interval treatment [38]. In addition, Gobbini et al., showed that patients who did not require the interval chemotherapy achieved better responses at rechallenge [17], which was further supported by a melanoma study in which patients who received targeted therapy as an interval treatment had inferior PFS when retreated [30]. The other independently reported predictive factors were discontinuation of primary ICI due to toxicity or clinical decision rather than PD, which is in line with our analysis [17], and a decrease in neutrophil-to-lymphocyte ratio after the intermediate treatment in spite of the increase during primary treatment [34]. Lastly, Wakasugi et al. showed that the superior OS was achieved with salvage ICI when compared to chemotherapy or radiotherapy [48]. Despite the promising efficacy of secondary ICI therapy, the treatment toxicity must also be addressed. Among analyzed studies, the prevalence of adverse events was relatively high and equal approximately 69% and 58% at primary and secondary ICI treatment, respectively. However, in the vast majority of cases these were easily manageable, as the frequency of severe adverse events were 21% and 16%, respectively. As such, ICI retreatment and rechallenge seem to be safe strategies for cancer treatment, which is in line with other reports focused on treatment toxicities [64,65,66].

## 5. Future Directions

To further investigate potential future directions of ICI retreatment and rechallenge, we looked at ongoing clinical trials reported in the Clinicaltrials.gov registry. Among phase I clinical trials, we found a phase Ib study of synergistic effect of rechallenge with PD-L1 inhibitor after PD-L1 immunotherapy (NCT05325684) and the study of synergistic response to rechallenge with G-CSF after prior anti-PD-1 treatment (NCT05222009). Further, among phase II clinical trials, we found the trial investigating novel compound Zimberelimab plus lenvatamib in advanced cervical cancer in patients who progressed on or after prior ICIs (NCT05824468), the study of Durvalumab in NSCLC patients who continue the treatment or are retreated with it (NCT04078152), the multi-center study of PD-1 inhibitor combined with hypofractionated radiotherapy and GM-CSF with IL-2 in patients with refractory to prior resistance to PD-1/PD-L1 advanced solid tumours (NCT05530200) and the study investigating retreatment with pembrolizumab after fecal microbiota transplantation in prostate cancer. Also, we found two phase III clinical trials on pembrolizumab rechallenge in melanoma with one of the arms for patients previously treated with pembrolizumab (NCT02362594) and a multimodal study investigating several ICIs such as: Pembrolizumab, Nivolumab, Atezolizumab, Durcalumab and Avelumab with an additional rechallenge arm (NCT04637594). Besides ICI therapies, we found several trials addressing retreatment or rechallenge strategies of targeted therapies (NCT01955681, NCT02514681, NCT00824746), PARP inhibitors (NCT05385068), chemotherapy (NCT00257114) or other treatment modalities. Despite the relatively low number of clinical trials, all of these studies will provide more data and should be followed. Nevertheless, there is a need to design new high-quality studies, taking into consideration the potential and challenges related to this treatment modality.

## 6. Conclusions

Evidence supporting all of these findings are limited, with multiple studies missing substantial data on the efficacy of primary and secondary treatment with ICIs. Many details of studies are unknown such as number of patients achieving certain RECIST responses at both ICI treatments, detailed course of drug administration, interval treatment, experienced toxicities and follow-up. Therefore, the specified data should be recorded in databases, which will enable more accurate analyses. Furthermore, the heterogeneity of cancer patients selected for original studies, including distinct prior and secondary oncological regimens, various administration schedules and lack of control cohorts or information on cessation from treatments, highly limits the ability of performing a relevant meta-analysis. More efforts should be made to standardize the treatment regimen between selected patients. All mentioned above must be addressed to effectively and safely translate ICIs into clinics in the nearest future. Thus far, the interval of chemoradiation application, primary response to the treatment as well as the cause of cessation to the first ICI are the strongest factors predicting good response. However, they are insufficient; hence, more studies to identify better predictors are needed.

## Figures and Tables

**Figure 1 cancers-15-03490-f001:**
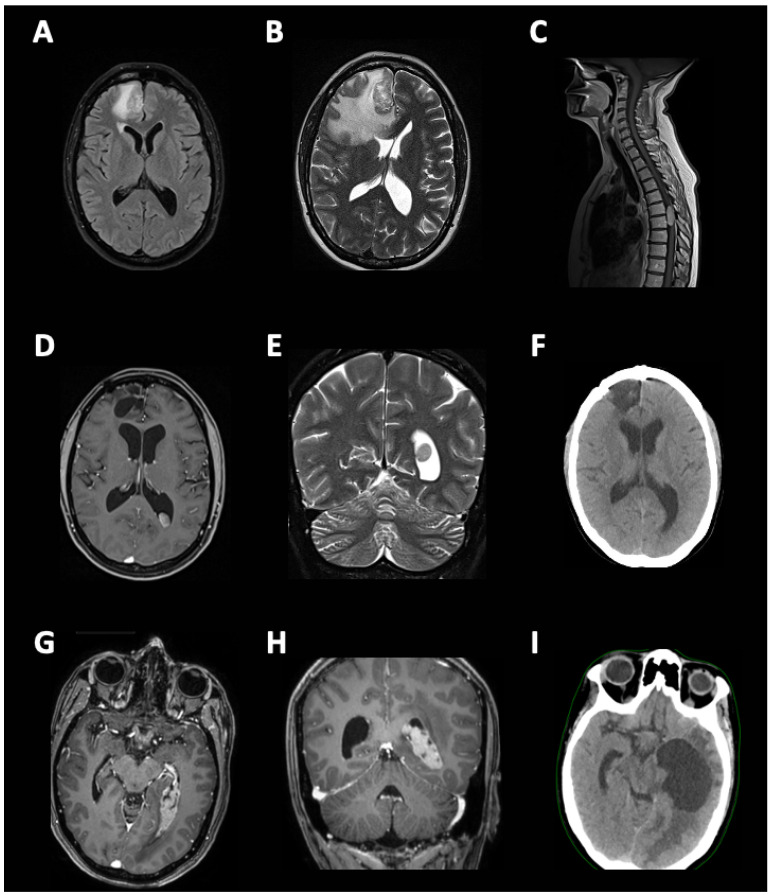
Central nervous system metastatic melanoma retreatment with pembrolizumab. The history of a 43-year-old female patient diagnosed with BRAF—negative skin melanoma (pT3a) of the right crus in March 2012. (**A**) April 2014; after multiple resections of locally recurrent lesions, head CT scan showed a lesion (33 mm × 23 mm × 29 mm) enhancing in post-contrast scan located in the right frontal lobe, which was removed via craniectomy during the same month and followed with adjuvant stereotactic radiotherapy (20 Gy/1 fraction, VMAT) in the following month. (**B**) August 2014; a new metastatic lesion in right frontal lobe, followed by surgical resection and 15 cycles of adjuvant dacarbazine chemotherapy between October 2014 and September 2015 (discontinued due to toxicity). (**C**) January 2017; disease relapse: new metastatic lesion in spinal canal (Th6-Th8) and skin lesions on lower (right/left—need to check) extremity. Neurosurgical resection of spinal tumor with excellent effect and low toxicity. (**D**,**E**) April 2017; diagnosis of metastatic lesion in posterior horn of the left lateral ventricle. Patient received stereotactic radiotherapy (20 Gy/1 fraction) and afterwards entered the clinical trial MasterKey-265 and received 34 doses of 200 mg of pembrolizumab in combination with local talimogene laherparepvec or placebo injection (April 2017–March 2019). (**F**) November 2017; central nervous system complete response achieved. (**G**,**H**) May 2021; central nervous system progression. Off-label 11 cycles of 200 mg of pembrolizumab was administered (June 2021–November 2021), achieving a partial response in September 2021. (**I**) January 2022; due to condition worsening since January, the patient received salvage, palliative radiotherapy which was not effective. The patient died in February 2022.

**Figure 2 cancers-15-03490-f002:**
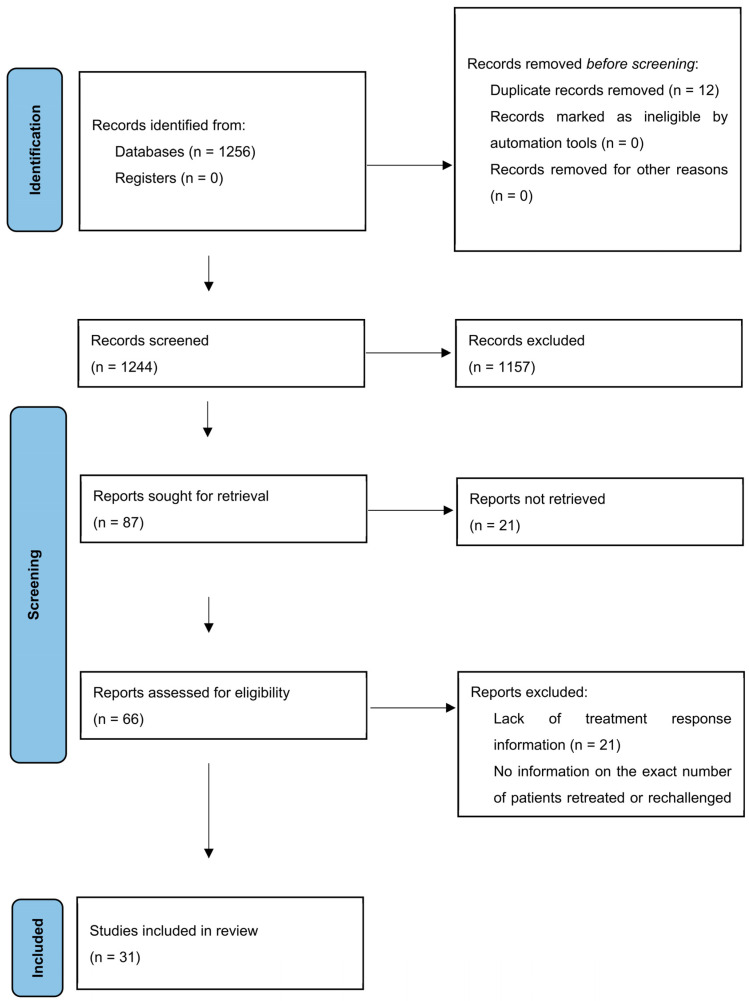
PRISMA Flowchart for systematic reviews.

**Table 1 cancers-15-03490-t001:** PICO-styled research question.

PICO	Description
Patients	Cancer patients that were subjected to immune checkpoint inhibitors retreatment or rechallenge
Indicator group	Cancer patients retreated or rechallenged with immune checkpoint inhibitors
Comparator group	Cancer patients who were not retreated or rechallenged with immune checkpoint inhibitors
Outcomes	Cancer patients’ RECIST-based response of immune checkpoint inhibitor retreatment or rechallenge

**Table 2 cancers-15-03490-t002:** Summary of analyzed studies.

First Author, Year	N	Stage	ICI1	CNV Meta	Interval Treatment	Median Time Interval	Cessation from ICI1	ICI2	CNV Meta	# Retreatment—1; # Rechallenge—2	Major Efficacy Outcome
	Melanoma										
Robert, 2013 [24]	38	IV	Ipi + gp100; Ipi	NA	NA	11.5 months (6.0–48.7); 8.9 months (6.0–28.9)	Completion	Ipi + gp100; Ipi	NA	1	7 patients achieved a better response after ICI2 than after ICI1.
Chiarion-Sileni, 2014 [25]	51	III–IV	Ipi	NA	NA	36 weeks (24–66)	Completion *	Ipi	3	1	Median OS in retreated vs. non-retreated was 21 (95% CI 16–26) and 13 months (95% CI 11–15), respectively (*p* < 0.0001).
Jacobsoone-Urlich, 2016 [26]	8	IV	Pem; Niv	NA	NA	127.5 days (91–210)	Completion *	Ipi	NA	Not-specified	4 patients achieved a better response after ICI2 than after ICI1.
Aya, 2016 [27]	9	IV	Pem; Niv	NA	4	13.1 weeks (2–38)	Progression	Ipi	1	Not-specified	2 patients who responded to ICI2, had DOR +8 and +17 months above median.
Bowyer, 2016 [28]	40	IV	Pem; Niv	NA	NA	53 days (range 2–683 days)	Progression	Ipi	3	2	ICI2 responders (>6 months OR or prolonged SD) achieved 3 PR, 3 SD and 1 PD at ICI1.
Nomura, 2017 [29]	8	IV	Niv	0	4	3 months	Progression (7); Toxicity (1)	Ipi	1	1	Median PFS in retreated vs. non-retreated was 4.1 (range 2.1–8.4) and 4.3 months (range 0.3–14), respectively.
Zimmer, 2017 [30]	84	III–IV	Anti-PD-1	NA	NA	42 days (1–588)—Niv + Ipi; 28 days (7–660)—Ipi	Progression *	Niv + Ipi; Ipi	33	1	Benefit from ICI1 had no impact on response to ICI2 (OR 0.75, 95% CI 2–3.6, *p* = 0.82; 1.45, 95% CI 4–5.0, *p* = 0.55).
Blasig, 2017 [31]	8	IV	Pem; Niv	0	6	10.5 months (1–15)	Progression (7); Toxicity (1)	Pem	3	1	2 patients achieved a better response after ICI2 than after ICI1.
Robert, 2019 [32]	13	III–IV	Pem	NA	3	NA	Competion (12); NA (1)	Pem	1	1	7 patients achieved the same ICI1 response after ICI2 (3 CR, 3 PR, 1 SD).
Jansen, 2019 [33]	19	III–IV	Pem; Niv	NA	NA	12 months (2.1–19.2)	Elective discontinuation	Pem; Niv	NA	1	5/6 objective responses after ICI2 were seen in patients obtaining CR during ICI1.
Kan, 2020 [34]	4	IV	Niv	0	4	NA	Completion	Pem	0	1	Patients who achieved PR at ICI2 showed a decrease in NLR after the intermediate treatment in spite of the increase during ICI1.
Whitman, 2020 [35]	21	III–IV	Pem; Niv	NA	NA	≥90 days	Completion (with SD or better)	Pem; Niv	0	1	Patients who achieved SD or better at ICI1 benefited from ICI2.
Hepner, 2021 [36]	47	III–IV	Ipi + Niv; Ipi + Pem; Ipi	NA	25	NA	Completion (27); Toxicity (19); Other (1)	Ipi + Niv; Ipi + Pem; Ipi	NA	1	Patients who achieved PR at ICI1 had a higher response rate to ICI2 than those with SD as the best response to ICI1 (11/33, 33% vs. 1/10, 10%, *p* = 0.035); however, PFS was similar
Perdyan, 2023 [Figure 1 of this paper]	1	IV	Pem	1	1	27 months	Completion	Pem	1	1	-
	Non-small cell lung cancer										
Niki, 2018 [37]	11	Advanced	Niv	NA	10	4.2 months (1–12.7)	NA (but not toxicity)	Niv; Pem	NA	2	4 patients who had responded to ICI1 responded to ICI2; the only patient who had PD at ICI1, achieved PR in ICI2, however, received chemoradiation in between.
Fujita, 2018 [38]	12	III–IV	Niv	NA	8	NA	Completion *	Pem	NA	1	Patients who responded to ICI2 (PR and SD) had very high (TPS ≥ 80%) tumour PD-L1 expression. Interval chemoradiation did not affect the efficacy of ICI2.
Fujita, 2019 [39]	18	III–IV	Pem; Niv	NA	11	NA	Completion (7); Progression (11) *	Atez	NA	1	Atezolizumab at ICI2 was not effective. It might be due to a large number of patients receiving it as a third- or later-line regimen.
Katayama, 2019 [40]	35	III–IV	Niv; Pem; Atez	NA	35 *	157 days (106–238)	Progression	Niv; Pem; Atez	7	2	In multivariate analysis, ECOG-PS ≥ 2 was associated with PFS (HR 2.38, 95% CI 1.03–5.52, *p* = 0.043) and OS (HR 3.01, 95% CI 1.10–8.24, *p* = 0.032) of ICI2.
Watanabe, 2019 [41]	14	III–IV	Pem; Niv; Atez	NA	14	6.5 months (2.1–15.1)	Progression	Niv; Pem	NA	2	2 of 3 patients who achieved more than SD at ICI2, received interval radiotherapy.
Mouri, 2019 [42]	21	III–IV	Niv	NA	0	NA	Toxicity	Niv	NA	1	Median OS and PFS did not differ between the ICI2 and discontinuation cohorts.
Gelsomino, 2020 [43]	1	III	Niv	0	1	13 months	Progression	Atez	0	2	-
Gobbini, 2020 [17]	144	I–IV	Anti-PD-1; Anti-PD-L1	24	88	NA	Progression (58); Toxicity (58); Clinical decision (28)	Anti-PD-1; Anti-PD-L1	33	2	Longer PFS and OS at ICI2 were achieved in cases of discontinuation of ICI1 because of toxicity or clinical decision (in most cases because of long-term benefit) compared to patients with PD. Moreover, patients who did not require an interval CHT, and those with a better ECOG PS at the ICI2 experienced better outcomes.
Xu, 2022 [44]	40	I–IV	Anti-PD-1	NA	7	NA	Progression	Anti-PD-1; Anti-PD-L1	10	2	Longer PFS was achieved in patients with the best overall response of SD/PD in initial immunotherapy, or whose treatment lines prior to ICI rechallenge were one or two.
	Breast cancer										
Otani, 2021 [45]	1	IV	Atez + Nab-p	NA	1	NA	Completion	Pem	NA	2	-
	Renal cell carcinoma										
Ravi, 2020 [46]	69	IV	Niv/Ipi-based	NA	NA	NA	Progression (50); Toxicity (16); Other (3)	Niv/Ipi-based	NA	2	The ORR at ICI-2 was higher in patients who responded to ICI-1 (7/24) compared with those who had SD (4/25) or PD (3/14), while it was similar in patients receiving single-agent ICI (n = 7), dual ICI (n = 5), or ICI in combination with TT (n = 3) at ICI-2.
	Mesothelioma										
Minchom, 2020 [16]	1	Advanced	Pem	NA	0	21 months	Completion	Pem	NA	2	-
	Classical Hodgkin lymphoma										
Kambhampati, 2022 [47]	7	III–IV	Avelumab	NA	4	3.2 months (0.5–23.7)	Progression (5); Toxicity (2)	Niv; Pem	NA	1	PD-1 blockade after PD-L1 blockade in r/r cHL may be effective with 86% ORR, including patients who previously progressed on avelumab.
	Head and neck cancer										
Wakasugi, 2022 [48]	12	NA	Niv; Pem	NA	12	NA	Progression	Niv	NA	1	In multivariate analysis, median OS was the longest in ICI2 when compared to salvage chemotherapy or radiotherapy cohorts (HR 0.258, 95% CI 091–0.732, *p* = 0.011).
	Urothelial carcinoma										
Makrakis, 2022 [49]	25	III–IV *	Anti-PD-1; Anti-PD-L1	0;1	13	45 weeks (8–208)	Progression (19); Toxicity (4); Completion (2) *	Anti-PD-1; Anti-PD-L1	0;1	2	About half of the patients who were rechallenged with an ICI-based regimen achieved disease control.
	Mixed cancers										
Martini, 2017 [50]	3	IV	Pem; Anti-PD-L1	0	3	6 months (0.25–20)	Progression (2); Toxicity (1)	Niv	0	1	-
Bernard-Tessier, 2018 [15]	8	NA	Anti-PD-1; Anti-PD-L1	NA	0	35 months (16.3–65.8)	Completion	Anti-PD-1; Anti-PD-L1	NA	2	Patients treated for MSI-high colorectal carcinoma and urothelial carcinoma had similar long-term responses to ICI2.
Simonaggio, 2019 [51]	40	NA	Anti PD-1; Anti PD-L1	NA	NA	NA	Toxicity	Anti PD-1; Anti-PD-L1	NA	2	Median PFS in rechallenged vs. non-rechallenged was 19.1 (95% CI 17—not reached) and 23.6 months (95% CI 10.2—not reached), respectively.

Legend: Atez—atezolizumab; CHT—chemotherapy; CI—confidence interval; DOR—duration of response; ECOG-PS—The Eastern Cooperative Oncology Group Performance Status; gp100—glycoprotein 100; HR—hazard ratio; ICI1—initial treatment with immune checkpoint inhibitors; ICI2—retreatment or rechallenge with immune checkpoint inhibitors; Ipi—ipilimumab; MSI—microsatellite instability; N—number of patients; NA—not available; Niv—nivolumab; NLR—neutrophils to lymphocyte ratio; OS—overall survival; OR—odds ratio; ORR—overall response rate; Pem—pembrolizumab; PD—progression disease; PD-1—programmed death receptor 1; PD-L1—programmed death ligand 1; PFS—progression-free survival; PR—partial response; r/r cHL—relapsed or refractory classical Hodgkin lymphoma; SD—stable disease; TPS—tumour proportion score; TT—targeted therapy; +—combination; /—or; *—lack of clear information; ^#^—according to authors.

**Table 3 cancers-15-03490-t003:** Detailed response description to initial and followed-up treatment with immune checkpoint inhibitors.

First Author, Year	N Patients	CR1	PR1	SD1	PD1	NE1	ORR1	DCR1	CR2	PR2	SD2	PD2	NE2	ORR2	DCR2
Melanoma															
Robert, 2013 [24]	38	0	11	21	5	1	0.29	0.84	1	6	16	15	0	0.18	0.61
Chiarion-Sileni, 2014 [25]	51	0	20	31	0	0	0.39	1	2	12	30	7	0	0.27	0.86
Jacobsoone-Urlich, 2016 [26]	8	0	2	0	6	0	0.25	0.25	3	1	0	4	0	0.5	0.5
Aya, 2016 [27]	9	0	4	1	4	0	0.44	0.56	0	2	0	7	0	0.29	0.29
Bowyer, 2016 [28]	40	0	8	15	17	0	0.2	0.58	0	4	3	33	0	0.1	0.18
Nomura, 2017 [29]	8	0	3	3	2	0	0.38	0.75	0	2	3	3	0	0.25	0.63
Zimmer, 2017 [30]	84	0	15	15	52	2	0.18	0.37	1	13	15	47	8	0.17	0.35
Blasig, 2017 [31]	8	1	2	3	2	0	0.38	0.75	0	1	3	4	0	0.13	0.25
Robert, 2019 [32]	13	6	6	1	0	0	0.92	0.92	3	4	3	0	2	0.54	0.77
Jansen, 2019 [33]	19	9	6	4	0	0	0.79	1	2	4	5	7	1	0.32	0.37
Kan, 2020 [34]	4	0	0	0	4	0	0	0	0	2	0	2	0	0.5	0.5
Whitman, 2020 [35]	21	4	10	7	0	0	0.67	1	7	6	5	3	0	0.62	0.86
Hepner, 2021 [36]	47	4	33	10	0	0	0.79	1	1	11	9	26	0	0.26	0.45
Perdyan, 2022 [Figure 1 of this paper]	1	1	0	0	0	0	-	-	0	1	0	0	0	-	-
Non-small cell lung cancer															
Niki, 2018 [37]	11	0	5	2	4	0	0.45	0.63	0	3	2	6	0	0.27	0.45
Fujita, 2018 [38]	12	0	7	2	3	0	0.58	0.75	0	1	4	6	1	0.08	0.42
Katayama, 2019 [40]	35	0	12	12	10	1	0.35	0.71	0	1	14	18	2	0.03	0.43
Watanabe, 2019 [41]	14	0	3	5	6	0	0.21	0.57	0	1	2	11	0	0.07	0.21
Fujita, 2019 [39]	18	0	7	4	6	1	0.39	0.61	0	0	7	11	0	0.39	0.39
Mouri, 2019 [42]	21	1	12	8	0	0	0.62	1	0	3	15	2	1	0.17	0.86
Gelsomino *, 2020 [43]	1	0	0	0	1	0	-	-	0	1	0	0	0	-	-
Gobbini, 2020 [17]	144	10	61	38	26	9	0.53	0.81	5	18	45	54	22	0.16	0.47
Xu, 2022 [44]	40	0	14	19	7	0	0.35	0.83	0	9	25	6	0	0.23	0.85
Breast cancer															
Otani *, 2021 [45]	1	0	1	0	0	0	-	-	1	0	0	0	0	-	-
Renal cell carcinoma															
Ravi, 2020 [46]	69	0	25	29	14	1	0.37	0.79	0	15	26	23	5	0.22	0.59
Mesothelioma															
Minchom *, 2020 [16]	1	0	1	0	0	0	-	-	0	0	1	0	0	-	-
Classical Hodgkin lymphoma															
Kambhampati, 2022 [47]	7	2	3	1	0	1	0.71	0.86	5	1	0	1	0	0.86	0.86
Head and neck cancer															
Wakasugi, 2022 [48]	12	1	1	8	2	0	0.17	0.83	0	2	5	1	1	0.17	0.58
Urothelial carcinoma															
Makrakis, 2022 [49]	25	3	6	4	11	1	0.36	0.52	1	3	8	12	1	0.16	0.48
Mixed cancers															
Martini, 2017 [50]	3	1	0	2	0	0	0.33	1	0	0	0	3	0	0	0
Bernard-Tessier, 2018 [15]	8	1	6	1	0	0	0.88	1	0	2	6	0	0	0.25	1
Simonaggio, 2019 [51]	40	0	9	17	4	10	0.3	0.87	0	13	15	9	3	0.33	0.7

Legend: CR—complete response; DCR—disease control rate; NE—not-evaluated; ORR—objective response rate; PD—progressive disease; PR—partial response; SD—stable disease; 1—initial treatment; 2—retreatment or rechallenge; * case report study.

**Table 4 cancers-15-03490-t004:** Progression-free survival and overall survival data.

First Author, Year	PFS1	PFS2	OS1	OS2
Melanoma				
Chiarion-Sileni, 2014 [25]	NA	NA	mOS: 21 months (95% CI 16–26)	mOS: 12 months (95% CI 10–14)
Jacobsoone-Urlich, 2016 [26]	NA	NA	NA	Mean OS: 13.8 months for 4 patients with CR or PR.
Aya, 2016 [27]	NA	mPFS: 3.14 months (95% CI 2.56–3.71)	mOS: 21.8 months (95% CI 12.9–30.6	mOS: 16.8 months (95% CI 8.1–25.4)
Bowyer, 2016 [28]	mPFS: 5 months (95% CI–not revealed)	NA	NA	NA
Nomura, 2017 [29]	mPFS: 4.1 months (range 2.1–8.4)	mPFS: 4.3 months (0.3–14)	mOS: 18.6 months (6.0–24.8)	NA
Zimmer, 2017 [30]	NA	mPFS: Ipi: 3 months (95% CI 2.8–3.8) Ipi + Niv: 2 months (95% CI 1.9–3)	NA	1-year OS: Ipi: 54% (95% CI 35–70) Ipi + Niv: 55% (95% CI 26–76)
Robert, 2019 [32]	mPFS: Pem: 11.6 months (95% CI 8.2–16.4) Ipi: 3.7 months (95% CI 2.8–4.3)	NA	mOS: Pem: 32.7 months (95% CI 24.5–41.6); Ipi: 15.9 months (95% CI 13.3–22)	NA
Jansen, 2019 [33]	mTTP: 12 months (2–23)	NA	NA	NA
Whitman, 2020 [35]	NA	NA	NA	mOS: 30 months (95% CI 14.4—not reached)
Hepner, 2021 [36]	mPFS: 11 months (95% CI 8–15)	mPFS: 5 months (95% CI 3–9)	NA	mOS: 17 months (95% CI 12—not reached)
Non-small cell lung cancer				
Niki, 2018 [37]	mPFS: 4.9 months (0.7–18.2)	mPFS: 2.7 months (0.5–16.1)	NA	NA
Fujita, 2018 [38]	mPFS: 6.2 months (range 2.8–13.7)	mPFS: 3.1 months (range 1.2–12.6)	NA	NA
Katayama, 2019 [40]	mPFS: 120 days (95% CI 84–139)	mPFS: 81 days (95% CI 41–112)	mOS: 596 days (95% CI 455–864)	mOS: 225 days (95% CI 106–361)
Watanabe, 2019 [41]	mPFS: 3.7 months (95% CI 1.3–7.1)	mPFS: 1.6 months (95% CI 0.8–2.6)	NA	OS: 6.5 months (95% CI 1.4–19.0)
Fujita, 2019 [39]	mPFS: Niv 7.7 months (±6.6) Pem 5.6 months (±4.7)	mPFS: 2.9 months (±1.8)	NA	NA
Gobbini, 2020 [17]	mPFS: 13 months (95% CI 10–16.5)	mPFS: 4.4 months (95% CI 3–6.5)	mOS: 3.3 years (95% CI 2.9–3.9)	mOS: 1.5 years (95% CI 1.0–2.1)
Xu, 2022 [44]	mPFS: 5.7 months (95% CI 4.1–7.2)	mPFS: 6.8 months (95% CI 5.8–7.8)	NA	NA
Renal cell carcinoma				
Ravi, 2020 [46]	mTTP: 8.2 months (95% CI 5.7–10.6)	mTTP: 5.7 months (95% CI 3.2–7.6)	NA	NA
Head and neck cancer				
Wakasugi, 2022 [48]	mPFS: 11.2 months (95% CI 0–29.3)	NA	mOS: 23.6 months (95% CI 21.1–26.0)	NA
Mixed cancers				
Bernard-Tessier, 2018 [15]	mPFS: 24.4 months (range 15.8–49.0)	mPFS: 12.9 months (range 5.0–35.4)	NA	NA
Simonaggio, 2019 [51]	mPFS: 19.1 (95% CI 17—not reached)	mPFS: 23.6 months (95% CI 10.2—not reached)	mOS: not reached	mOS: not reached

Legend: CI—confidence interval; CR—complete response; Ipi—ipilimumab; m—median; NA—not available; Niv—nivolumab; OS1—overall survival of initial treatment; OS2—overall survival of retreatment or rechallenge; PFS1—progression-free survival of initial treatment; PFS2—progression-free survival of retreatment or rechallenge; TTP—time to progression.

**Table 5 cancers-15-03490-t005:** Toxicities of immune checkpoint inhibitors at initial and followed-up treatment.

First Author, Year	N	AE1 (%)	AE2 (%)	SAE1 (%)	SAE2 (%)
Melanoma					
Robert, 2013 [24]	38	0.47	0.58	NA	NA
Chiarion-Sileni, 2014 [25]	51	0.39	0.22	0.04	0.06
Jacobsoone-Urlich, 2016 [26]	8	NA	0.38	NA	0.13
Aya, 2016 [27]	9	0.67	0.88	0.11	0.44
Bowyer, 2016 [28]	40	NA	NA	0.21	0.35
Nomura, 2017 [29]	8	1.25	0.38	0.25	0
Blasig, 2017 [31]	8	1.63	1.38	0.25	0.13
Robert, 2019 [32]	13	NA	0.46	NA	0
Kan, 2020 [34]	4	0.25	0.5	0	0
Hepner, 2021 [36]	47	0.94	0.57	0.38	0.38
Non-small cell lung cancer					
Niki, 2018 [37]	11	1.45	0.64	0	0
Fujita, 2018 [38]	12	1.25	1.33	0.67 (G ≥ 2)	0.83 (G ≥ 2)
Watanabe, 2019 [41]	14	0.64	0.36	0.21 (G ≥ 2)	0 (G ≥ 2)
Fujita, 2019 [39]	18	0.89	1.28	0.5 (G ≥ 2)	0.83 (G ≥ 2)
Mouri, 2019 [42]	21	NA	1	NA	0.33
Gobbini, 2020 [17]	144	NA	NA	0.19	0.06
Xu, 2022 [44]	40	NA	NA	NA	NA
Renal cell carcinoma					
Ravi, 2020 [46]	69	0.71	0.45	0.26	0.16
Classical Hodgkin lymphoma					
Kambhampati, 2022 [47]	7	0.43	0.14	0.14	0
Head and neck cancer					
Wakasugi, 2022 [48]	12	NA	1.5	NA	0
Urothelial carcinoma					
Makrakis, 2022 [49]	25	0.17	0	NA	NA
Mixed cancers					
Martini, 2017 [50]	3	NA	1	NA	NA
Simonaggio, 2019 [51]	40	NA	1	NA	0.55

Legend: AE1—adverse effects at initial treatment; AE2—adverse effects at retreatment or rechallenge; G—grade; ICI1—initial treatment with immune checkpoint inhibitors; ICI2—retreatment or rechallenge with immune checkpoint inhibitors; Ipi—ipilimumab; N—number of patients; NA—not available; Niv—nivolumab; Pem—pembrolizumab; SAE1-—severe adverse effects at initial treatment; SAE2—severe adverse effects at retreatment or rechallenge; AE and SAE defined as number of events, not patients (SAE = grade ≥ 3).

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
