# Peer review of "The Effectiveness of Cancer Immune Checkpoint Inhibitor Retreatment and Rechallenge—A Systematic Review"

_cancers, 2023, doi:10.3390/cancers15133490_

Round 1

Reviewer 1 Report

The systematic review titled “The effectiveness of cancer immune checkpoint inhibitors retreatment and rechallenge. A systematic review by Perdyan et al. is very well written. The authors designed the manuscript excellently and made a conclusion on the basis of major evidence. Overall, the quality of the manuscript is very good. However, the manuscript needs revision and my comments are as follows.

Comment 1. What is the importance of this systematic review? Kindly elaborate in the introduction section.  

Comment 2. What are the challenges ahead? What are the challenges in the clinical translation of immune checkpoint inhibitors?

Comment 3. The authors should also discuss about the patents and clinical products in a separate section.

Author Response

Thank you for your review and comments. We highly appreciate your help.

Comment 1. What is the importance of this systematic review? Kindly elaborate in the introduction section.  / This comment was addressed in the introduction section.

Comment 2. What are the challenges ahead? What are the challenges in the clinical translation of immune checkpoint inhibitors? / This comment was addressed in the conclusions section.

Comment 3. The authors should also discuss about the patents and clinical products in a separate section. / We added Future Directions section at the end of manuscript discussing ongoing clinical trials.

Reviewer 2 Report

see attached file

Author Response

Thank you for your review and comments. We highly appreciate your help.

Answers:

In my opinion, a very important finding that the Authors did not report is the cause of cessation of the first treatment (completion of planned therapy, patient refusal, toxicity, progression, other). This is very important because it affects the effectiveness of retreatment. Therefore, the Authors are invited to provide this data.

Information about cessation of the first treatment was added to Table 2 and further discussed within results and discussion sections.

Discussion

The sentence reported by the Authors “Hence, monotherapy with nivolumab in primary setting followed by pembrolizumab seems to be justified” is not supported by literature, neither the biochemical differences support a different clinical efficacy of the two drugs, therefore the phrase must be removed.

  1. The similar efficacy of the two drugs is well known.

It is also known that the clinical efficacy of the two drugs is completely comparable even if there are no randomized direct comparison studies.

The sentence was removed.

References

Please, add the following recent papers from our group in introduction:

De Risi I, Sciacovelli AM, Guida M. Checkpoint Inhibitors Immunotherapy in Metastatic Melanoma: When to Stop Treatment? Biomedicines. 2022 Sep 28;10(10):2424. doi: 10.3390/biomedicines10102424. PMID: 36289687; PMCID: PMC9599026.

The reference was added in the discussion.

Reviewer 3 Report

I have reviewed the manuscript entitled "The effectiveness of cancer immune checkpoint inhibitors pretreatment and rechallenge by Perdyan et al.  

The manuscript represents a global overview of the effectiveness of checkpoint inhibitor re-treatment across a broad spectrum of tumor types based on a review of existing literature.  This represents a useful and novel overview of a developing challenge (patients who relapse after prior response to checkpoint inhibitor treatment).

I believe the methods and interpretation of the data are appropriate.

I have several concerns:

1) Due to the nature of the retrospective review, there is significant heterogeneity of the included patients.  In some articles, patients had treatment discontinued with stable disease or partial responses.  In others, patients had achieved complete remissions.  It should be pointed out that there are potentially significant differences between patients who never achieved a complete remission and have a relatively brief progression-free interval prior to pretreatment versus patients with lengthy initial complete remission with late relapses. I suspect the biology responsible for progression may be different in these populations.  Thus additional prospective studies of systematic and identical re-treatment approaches in these patient populations are recommended.

There also may be significant difference in the effectiveness of retreatment with single agent CTLA4 and PD-1/PDL1 antibodies versus combination therapies (e.g. CTLA4+PD1 antibodies, targeted agent and checkpoint inhibitor combinations).  Thus not all of the studies evaluated have equivalent significance. There was not an attempt to separately analyze patients treated with more active compared to less active retreatment strategies.

The main value to this global review is to point out the developing challenge of rare patients who relapse or progress after apparent clinical remission (which can occur many years after treatment cessation).  It is clear from this review that some of these patients may have another significant response to retreatment.  This needs to better characterized in each tumor type and re-treatment regimens optimized.  

Author Response

Thank you for your review and comments. We highly appreciate your help.

Answers:

1) Due to the nature of the retrospective review, there is significant heterogeneity of the included patients.  In some articles, patients had treatment discontinued with stable disease or partial responses.  In others, patients had achieved complete remissions. / In our article, we highlighted the heterogeneity between all included studies gathering information on treatment responses.

It should be pointed out that there are potentially significant differences between patients who never achieved a complete remission and have a relatively brief progression-free interval prior to pretreatment versus patients with lengthy initial complete remission with late relapses. I suspect the biology responsible for progression may be different in these populations.  Thus additional prospective studies of systematic and identical re-treatment approaches in these patient populations are recommended. / We added that information in the discussion.

There also may be significant difference in the effectiveness of retreatment with single agent CTLA4 and PD-1/PDL1 antibodies versus combination therapies (e.g. CTLA4+PD1 antibodies, targeted agent and checkpoint inhibitor combinations).  Thus not all of the studies evaluated have equivalent significance. There was not an attempt to separately analyze patients treated with more active compared to less active retreatment strategies. / It is a valuable point into the discussion. However, almost in all studies patients were treated with single agents which makes such analysis beyond the scope of this study.

The main value to this global review is to point out the developing challenge of rare patients who relapse or progress after apparent clinical remission (which can occur many years after treatment cessation).  It is clear from this review that some of these patients may have another significant response to retreatment.  This needs to better characterized in each tumor type and re-treatment regimens optimized. / In depth analysis of biology of each patients is impossible due to lack of such data among included studies which is pointed out in the article. Indeed, our work highlights all available data and also points out data which should have been reported before in order to run more complex analysis. Additionally, besides melanoma and NSCLC, we have access only to single studies with regard to other cancer types. We hope that in the future such analysis will be possible. We added a future perspective paragraphs pointing out all ongoing research in the topic which should be followed for future considerations.  

Round 2

Reviewer 1 Report

The authors revised the manuscript very carefully. The revision is satisfactory.

Reviewer 2 Report

The paper is acceptable in its the current form